# Estimating soil fungal abundance and diversity at a macroecological scale with deep learning spectrotransfer functions

Yuanyuan Yang[1], Zefang Shen[1], Andrew  Bisset[2], and Raphael A. Viscarra Rossel[1]

[1]Soil and Landscape Science, School of Molecular and Life Sciences, Curtin University, GPO Box U1987, Perth WA 6845, Australia.
[2]CSIRO Oceans and Atmosphere, GPO BOX 1538, Hobart TAS 7001, Australia.

**Correspondence:** Raphael A. Viscarra Rossel (r.viscarra-rossel@curtin.edu.au)

**Abstract.** Soil fungi play important roles in the functioning of ecosystems, but they are challenging to measure. Using a continental scale dataset, we developed and evaluated a new method to estimate the relative abundance of the dominant phyla and diversity of fungi in Australian soil. The method relies on the development of spectro-transfer functions with state-of-the-art machine learning and using publicly available data on soil and environmental proxies for edaphic, climatic, biotic and topographic factors, and visible–near infrared (vis–NIR) wavelengths, to estimate the relative abundances of the Ascomycota, Basidiomycota, Glomeromycota, Mortierellomycota and Mucoromycota and community diversity measured with the abundance-based coverage estimator (ACE) index. The machine learning algorithms tested were partial least squares regression (PLSR), random forest (RF), Cubist, support vector machines (SVM), Gaussian process regression (GPR), XG-boost (XGB) and one-dimensional convolutional neural networks (1D-CNNs). The spectro-transfer functions were validated with a 10-fold cross-validation (n = 577). The 1D-CNNs outperformed the other algorithms and could explain between 45 and 73 % of fungal relative abundance and diversity. The models were interpretable, and showed that soil nutrients, pH, bulk density, an ecosystem water balance (a proxy for aridity) and net primary productivity were important predictors, as were specific vis–NIR wavelengths that correspond to organic functional groups, iron oxide and clay minerals. Estimates of the relative abundance for Mortierellomycota and Mucoromycota produced $R^2 \geq 0.60$, while estimates of the abundance of the Ascomycota and Basidiomycota produced $R^2$ values of 0.5 and 0.58, respectively. The spectro-transfer functions for the Glomeromycota and diversity were the poorest with $R^2$ values of 0.48 and 0.45, respectively. There is no doubt that the method provides estimates that are less accurate than more direct measurements with conventional molecular approaches. However, once the spectro-transfer functions are developed, they can be used with very little cost, and could serve to supplement the more expensive and laborious molecular approaches for a better understanding of soil fungal abundance and diversity under different agronomic and ecological settings.

# 1 Introduction

Soil fungi are important components of microbial communities, which inhabit dynamic soil environments. They play critical functional roles as decomposers, mutualists, and pathogens (Li et al., 2019). They impact nutrient cycling and ecosystem services, such as soil carbon fixation, fertility and productivity (Vetrovsky et al., 2019; Delgadobaquerizo et al., 2016). Given the important functions that soil fungi perform, it is important to better characterise and understand their communities over large scales. However, data on soil fungi are few or largely unavailable because the measurement of soil fungi, which needs field sampling, followed by culture-based analysis or DNA sequencing, are laborious, time-consuming and costly. Using soil sensing technologies, such as spectroscopy together with molecular approaches could greatly improve the utility of fungal inventory data (Hart et al., 2020).

Improvements in soil analytical methodologies provide an opportunity to increase sampling density for deriving a more detailed understanding of soil properties, their spatial variation, soil condition, and to improve decision-making. Spectroscopic techniques, such as visible–near infrared (vis–NIR) spectroscopy, have been developed to provide rapid estimates of soil properties (Viscarra Rossel et al., 2016). Soil vis–NIR spectra are largely nonspecific because of the overlapping absorptions of soil constituents (Stenberg et al., 2010). Complex absorption patterns generated from soil constituents need to be mathematically extracted from the spectra and there are various methods that can be used to model soil properties with spectra. They include multivariate statistical methods such as partial least squares regression (PLSR), and machine learning with different algorithms, including neural networks (Viscarra Rossel and Behrens, 2010; Morellos et al., 2016; Liu et al., 2018; Tsakiridis et al., 2020; Shen and Viscarra Rossel, 2021). Thus, vis–NIR spectra can integrally characterize the soil's mineral-organic composition, and combined with multivariate modelling, soil spectroscopy provides a rapid and cost-efficient method for soil characterisation (Viscarra Rossel and Brus, 2018).

Although there are no vis–NIR absorptions that can be directly assigned to soil microbial communities or diversity, soil microbes are dependent on the fundamental soil composition: its minerals, organic matter and water content. For example, they rely on organic matter for energy, on clay minerals and iron oxides for the supply of essential elements to grow (Müller, 2015). These organic and mineral properties are well represented and have a direct response in soil vis–NIR spectra (Stenberg et al., 2010). Therefore, vis–NIR spectra have been used to model various functional soil properties, such as soil organic carbon, cation exchange capacity, pH, clay content (Shi et al., 2015), as well as soil microbial communities (Davinic et al., 2012; Yang et al., 2019). For the latter, if the microbial biomass is present in the soil organic matter, then the spectra might well detect their functional constituents.

There are studies that use environmental proxies, (or covariates) at continental and global scales to model soil microbial properties using various methods, including linear regressions and machine learning (Serna-Chavez et al., 2013; Griffiths et al., 2011; Vetrovsky et al., 2019; Yang et al., 2019; Delgadobaquerizo et al., 2018a). However, we found no published studies that used vis–NIR spectra or a combination of spectra with other soil and environmental covariates (i.e. spectro-transfer functions) to infer fungal abundance or diversity. In a previous study, Yang et al. (2019) showed that vis–NIR spectra combined with other soil and environmental data could estimate soil bacterial abundance and diversity. Here, our hypotheses are: (i) spectroscopic

models with machine learning can estimate soil fungal abundance and diversity at the continental scale, and (ii) spectro-transfer functions with additional predictors to capture other soil and environmental properties that affect soil fungi will improve the accuracy of the estimates.

Thus, our objective is to develop and test the spectroscopic method for estimating soil fungal abundance and diversity over a large scale, and our aims are to:

(i) Compare the modelling of fungal abundance and diversity with vis–NIR spectra only (spectroscopic models), with readily available soil and environmental data only (environmental models) and with the combined set of vis–NIR spectra and readily available soil and environmental data (spectro-transfer functions), and

(ii) Test different statistical and machine learning algorithms for the modelling.

## 2 Methods

### 2.1 Soil sampling and laboratory analyses

We used 577 soil samples from the Biomes of Australian Soil Environments (BASE) project (Bissett et al., 2016). In that project, sampling were undertaken from soil that supports diverse plant communities across Australia. The sampling was carried out during the growing season when hydrothermal conditions are most conducive to typical plant growth. In the higher rainfall forested regions of the continent, the soil samples were collected mostly in spring and summer from September to February. In the shrublands and grasslands of the semi-arid and arid interior, soil samples were collected in spring from September to November. In the transitional zone between the southeast coast and the more arid interior, soil samples were collected in mainly autumn from March to May. Samples came from two soil depths (0–0.1m and 0.2–0.3m), covering five typical Australian ecosystem types, including cropland, forest, grassland, shrubland, and woodland (Fig. 1a). Woodlands in Australia represent ecosystems which contain widely spaced trees, the crowns of which do not touch. Woodlands consist of areas with fewer and more scattered trees than forests. In temperate Australia, woodlands are mainly dominated by Eucalyptus species. Temperate woodlands occur predominantly in regions with a mean annual rainfall of between 250–800mm, forming a transitional zone between the higher rainfall forested margins of the continent and the shrub and grasslands of the arid interior. Each sample was partitioned into subsamples for DNA sequencing (see below) and air-dried and crushed to a particle size of ≤2 mm for physicochemical analyses. The soil properties analyzed were total organic carbon and soil nutrients (e.g. ammonium, nitrate, phosphorus, potassium), pH, exchangeable cations (aluminium, sodium, magnesium, calcium), and texture (sand, silt and clay). The methods are described in (Bissett et al., 2016). Subsamples of the ≤2 mm portions were used for the spectroscopic analysis (see below).

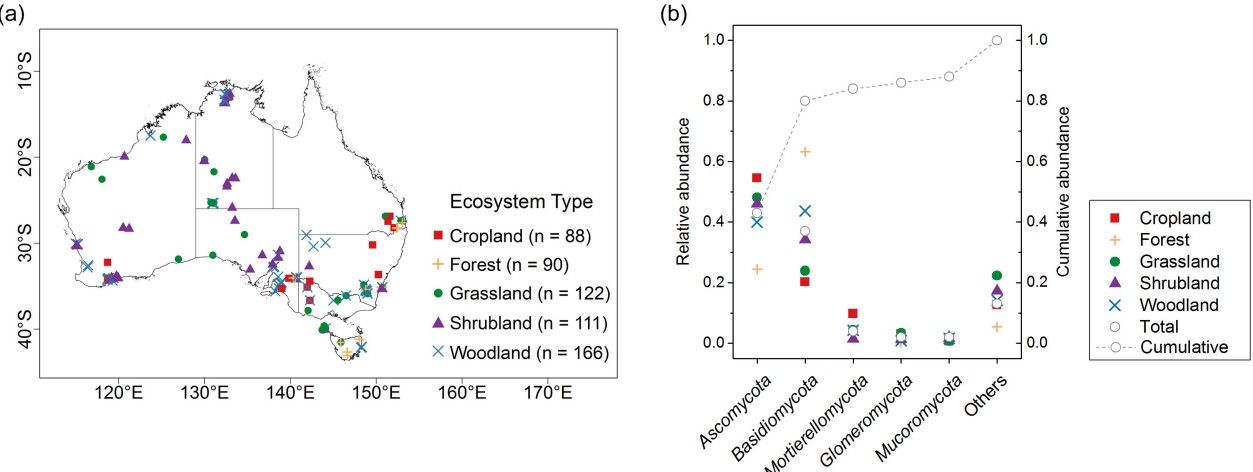

**Figure 1.** (a) Sampling sites and the range of ecosystem types across Australia (b) The mean relative abundances of dominant fungal phyla and unclassified "Others" taxa in five ecosystem types. Individual abundance of each phylum and their cumulative abundance were shown in the graph.

## 2.2 Derivation of fungal abundance and diversity

The methods for DNA extraction and sequencing are detailed in Bissett et al. (2016). Briefly, the soil DNA was extracted in triplicate following methods used in the Earth Microbiome Project[1]. Sequencing occurred with an Illumina MiSEQ, which is described in the BASE protocols[2]. Summarising, amplicons targeting the fungal ITS region were prepared and sequenced for each sample. The ITS amplicons were sequenced using 300 bp paired end sequencing. ITS1 regions were extracted using ITSx Bengtsson-Palme et al. (2013). Sequences comprising full and partial ITS1 regions were passed to the Operational

Taxonomic Units (OTU) selection and assigning workflow Bissett et al. (2016), which followed guidelines described in the BASE protocols[3] and in Bissett et al. (2016). These are based on the most current version of UNITE database (version 8.2, updated 15-01-2020) for molecular identification of fungi Nilsson et al. (2018). We used the final sample-by-OTU data matrix and annotated taxonomy file for the analyses of fungal diversity and composition.

To eliminate bias on the diversity comparison caused by unbalanced sequencing, samples were resampled at the same

sequencing depth using functions of the RAM library in the R software(R Core Team, 2014). The BASE dataset sought to produce as many sequences as resources allow with a minimum sequencing number of 10,000 per sample. Here, 11 000 sequences (the median number of sequences in the samples) were used as the resampling depth, because the majority of samples only had this amount of sequences, but also because the rarefaction curves started to flatten out for all 577 samples at this sequencing depth. This suggested that the sequencing number was sufficient (Fig. S2 in the Supplementary Information).

---

[1]http://www.Earthmicrobiome.Org/emp-standard-protocols/dna-extraction-protocol/

[2]https://ccgapps.Com.Au/bpa-metadata/base/information

[3]https://ccgapps.com.au/bpa-metadata/base/information

To quantify community diversity, we then calculated the abundance-based coverage estimator (ACE) index (Lozupone and Knight, 2008) from the resampled sample-by-OTU matrix. The relative abundance of fungal phyla were then determined using the ratio of sequences number classified at each phylum to the total number of sequences of each sample.

## 2.3 Soil visible–near-infrared spectroscopy

We measured the diffuse reflectance spectra of all air-dried $\leq$ 2 mm soil samples with the Labspec® vis–NIR spectrometer
(Malvern Panalytical, Boulder, Colorado, USA) following the protocols described in Viscarra Rossel et al. (2016). The spectral range of the spectrometer is 350 to 2500 nm. Due to the low signal-to-noise ratio at the start and end of each spectrum, for our analysis, we kept only spectra in the range between 380 and 2450 nm. As the spectra are highly collinear, to reduce redundancy in the data, we re-sampled them to a resolution of 10 nm. The measurements were performed with the instruments high intensity contact probe (PaNalytic, Boulder, Colorado, USA), and a Spectralon® white reference panel was used for calibration once
every 10 measurements.

For the modelling and interpretation, we first transformed the reflectance (R) spectra to apparent absorbance, using $A = \log_{10}(1/R)$, and then used the Savitzky-Golay method with a window of size 7, a quadratic polynomial and first derivative Savitzky and Golay (1964) to remove baseline effects and to improve the signal-to-noise ratio. To visualise the spectra, we further fitted each reflectance (R) spectrum with a convex hull and computed the deviations from the hull (Clark and Roush, 1984). These
115 continuum removed (CR) spectra help to visualise the characteristic absorptions, more clearly than the Savitzky-Golay first derivatives (SG1Der) absorbance spectra.

## 2.4 Modelling soil fungal abundance and diversity

We developed spectroscopic models, environmental models, and spectro-transfer functions for estimating soil fungal abundance and diversity (see below). The spectroscopic models used only the vis–NIR spectra, the environmental models used only the
120 publicly available soil and environmental data that represent the soil forming factors soil, climate, vegetation, terrain and parent material (Jenny, 1994), and the spectro-transfer functions used the vis—NIR spectra together with soil and environmental data.

We assembled a set of readily available soil and environmental maps that represented climate, terrain, vegetation, and parent material. To relate the these covariates to the fungal data, we extracted values from these maps using the geographic coordinates of the sample set. The soil property data came from the Australia-wide fine spatial resolution ($90 \times 90$ m) digital soil maps of
125 total organic carbon, total nitrogen, total phosphorus, bulk density, effective cation exchange capacity, available water capacity, pH, and soil texture (sand, silt, and clay) (Viscarra Rossel et al., 2015), as well as maps of the clay minerals kaolinite, illite, and smectite (Viscarra Rossel, 2011). To represent climate, we used data on mean annual temperature (MAT), mean annual precipitation (MAP), solar radiation, and evapotranspiration (Xu and Hutchinson, 2011) and the Prescott index (PI) (Prescott, 1950), which is calculated as the ratio of precipitation to evapotranspiration. To capture functional landscape characteristics,
we used a digital elevation model (DEM) from the 3-arc second shuttle radar topographic mission (SRTM) and derived terrain attributes (Gallant et al., 2011). To represent vegetation, we used data on net primary productivity (NPP) (Haverd et al., 2013), and on the fraction of photosynthetically active radiation intercepted by the sunlit canopy of the evergreen (Fpar-e) and woody

(Fpar-r) vegetation (Donohue et al., 2009). To represent parent material, we used gamma radiometrics, which comprises data on potassium, uranium, and thorium (Minty et al., 2009). Supplementary Table S2 lists these data and their main characteristics.

The spectra and the covariates were centred and scaled before the modelling of fungal abundance and diversity. The algorithms that we tested were partial least squares regression (PLSR) (Wold et al., 2001), gaussian process regression(GPR) (Rasmussen and Williams, 2005), support vector machines (SVM) (Suykens et al., 2002), random forest(RF) (Breiman, 2001), CUBIST (Quinlan, 1992), extreme gradient boost (XGBoost) (Friedman, 2001) and optimised 1D convolutional neural networks (1D-CNNs) (Shen and Viscarra Rossel, 2021). The algorithms and their implementation are described in the Supplementary Information linked to this article.

The predictability of the spectroscopic models and the spectro-transfer functions were assessed using 10-fold cross-validations. We evaluated the estimates using the Nash Sutcliffe model efficiency, other wise known as the coefficient of determination ($R^2$), which represent the fraction of the explained variance based on the 1:1 line of estimated versus measured values (Janssen and Heuberger, 1995). The $R^2$ was computed as 1-RSS/TSS, where RSS is the residual sum of squares and TSS is the total sum of squares. The root mean squared error (RMSE), which measures inaccuracy, the standard deviation of the error (SDE), which measures imprecision and the mean error (ME), which measures bias (Viscarra Rossel and McBratney, 1998). Inaccuracy (RMSE) embraces both the bias (ME) and the imprecision (SDE) (Viscarra Rossel and McBratney, 1998). Their relationship is given by $\mathrm{RMSE}^2 = \mathrm{ME}^2 + \mathrm{SDE}^2$.

To interpret the models, we calculated their variable importance as follows. For the PLSR, GPR, SVM, Cubist, RF and XGBoost models, variable importance was calculated using the varImp function in the caret library (Kuhn et al., 2008) of the software R. To calculate the variable importance of the CNN models, we used permutation variable importance. In our case, we run 1000 permutations and measured the decrease in RMSE after a predictor was permuted (randomly rearranged). The permutation breaks the relationship between the predictor and the response variables, and a reduction in RMSE indicates how much the model depends on the particular predictor. An advantage of this approach is that it can be applied on any estimator and does not require retraining the model (Breiman, 2001; Fisher et al., 2019). In order to compare the importance between different fungal phyla and diversity, we scaled the importance values between 0 and 1. In the results, we only report the variable importance of the model that performed best.

## 3   Results

In total, more than 60 million quality filtered sequences in the whole dataset were obtained, with an average of 107 310 sequences per sample. When we clustered the sequences at 97% similarity level 202 200 OTUs were detected. Each sample had an average of 666 OTUs. Sixteen phyla were identified in total and 5 dominant phyla, with relative abundance > 2%, were approximately present in most soils. This represented nearly 88% of the sequence number. The relative abundance of fungal phyla varied across ecosystem types (Fig. 1b).

Ascomycota (mean 0.43, SD 0.21) was the most abundant phylum, followed by Basidiomycota (mean 0.37, SD 0.24) (Table 1). Dominant fungal phyla showed a high degree of variability, with an averaging 83% coefficient of variation (CV). The

ACE index showed a wide range from 81 to 1823 (mean 563, SD 315). The rich soil biodiversity of the data resulted from the extensive soil sampling taken from diverse vegetation, soils, and climates across Australia.

**Table 1.** The descriptive statistics of relative abundance of dominant phyla and community diversity (n = 577).

| Variables | Mean | Median | St. Dev. | Range | Coeff. var. (%) |
|---|---|---|---|---|---|
| Abundance | | | | | |
| *Ascomycota* | 0.43 | 0.42 | 0.21 | 0.04–0.98 | 49 |
| *Basidiomycota* | 0.37 | 0.32 | 0.24 | 0.01–0.92 | 65 |
| *Mortierellomycota* | 0.04 | 0.02 | 0.04 | 0.00–0.36 | 100 |
| *Glomeromycota* | 0.02 | 0.01 | 0.01 | 0.00–0.41 | 50 |
| *Mucoromycota* | 0.02 | 0.01 | 0.03 | 0.00–0.55 | 150 |
| Diversity | | | | | |
| ACE | 563 | 503 | 315 | 81–1823 | 56 |

Fig. 2 shows the CR reflectance and SG1Der absorbance spectra with the characteristic absorption features. Soil with different fungal diversity show variations in absorptions, particularly around those that are due to Fe-oxides (400–800 nm), minerals (around 1400 nm, 1900 nm and 2200 nm) and organic compounds (throughout the vis–NIR spectrum) (Stenberg et al., 2010). Soil with the lower fungal diversity showed a more pronounced absorbance around 600 nm as shown in Fig. 2. In our study, the soil with lower fungal diversity mainly come from the central and western Australia. In these areas, soil subjected to intense weathering regimes and can accumulate large quantities Fe oxides (total soil $Fe_2O_3$ larger than 10%) in surficial environments, and strongly absorbed in the visible region (Viscarra Rossel et al., 2010). These highly iron-rich lateritic soil occur with acidic pH, high $H_2O$ and Al activities, and has been shown not conductive to the development of fungal diversity (Viscarra Rossel et al., 2010).

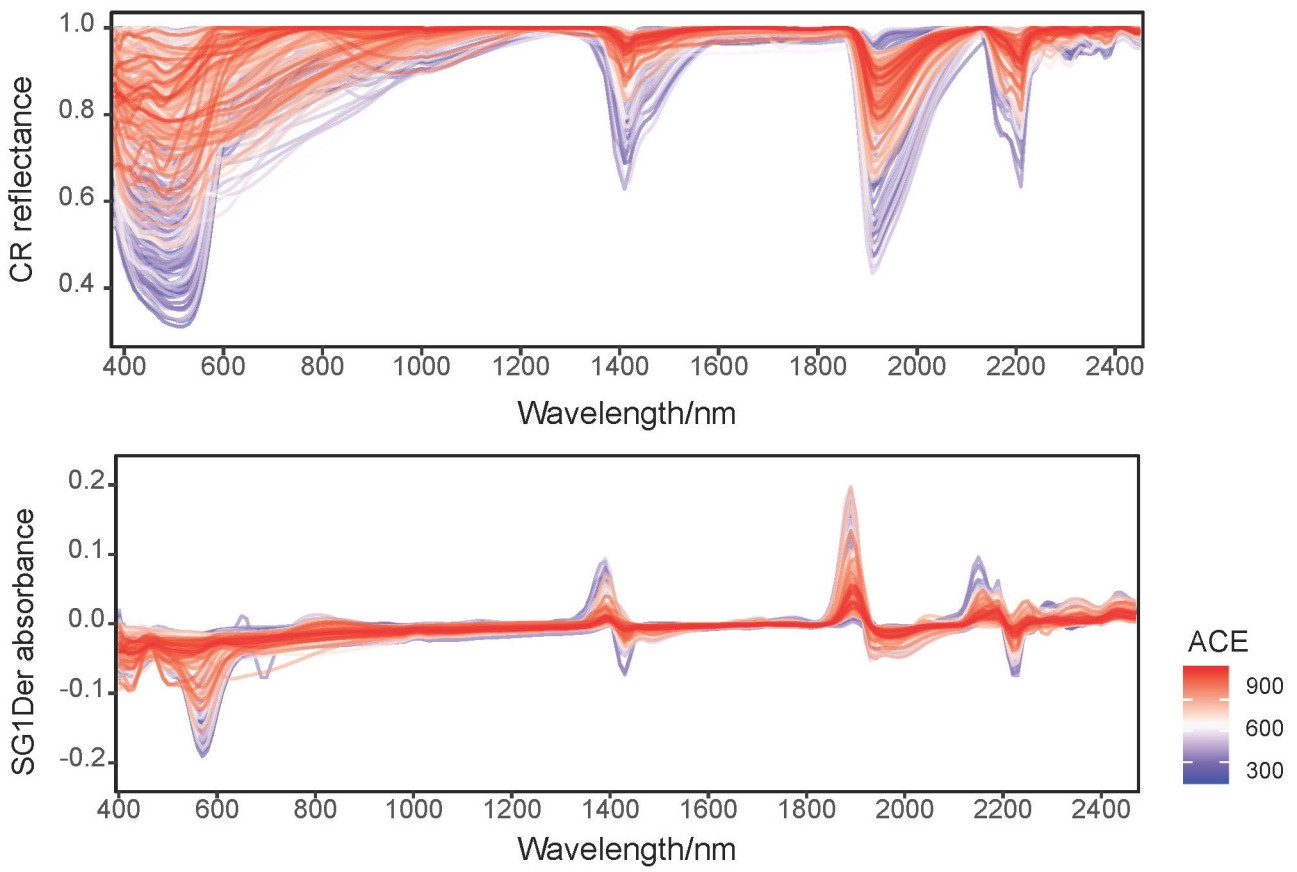

**Figure 2.** Continuum removed (CR) spectra and the Savitzky-Golay first derivatives (SG1Der) absorbance spectra curves colored by fungal ACE diversity

### 3.1 Modelling

With the different algorithms, the spectroscopic models (i.e. with only the vis–NIR spectra) could explain 9–45% of the variation in fungal phyla relative abundance and diversity. Spectroscopic models of the Glomeromycota were the least successful, with $R^2$ values ranging from 0.09 using SVM to 0.30 using 1D-CNN, while those of the Mortierellomycota produced the largest $R^2$ values, ranging from 0.32 using XGBoost to 0.45 using 1D-CNN (Fig. 3). The models of diversity had $R^2$ values ranging from 0.14 with PLSR to 0.35 using 1D-CNN.

The models derived with the readily available soil and environment data could explain 14–60% of the variation in fungal phyla relative abundance and diversity with the different algorithms. These environmental models were generally better performed than spectroscopic models, with an average 10% additional variance explained.

Combining the vis–NIR spectra and soil and environmental data further improved the modelling and their explanatory power. The spectro-transfer functions (i.e. with the combined set of vis–NIR spectra and other soil and environmental data) performed,

on average, 20% better than the spectroscopic models and 10% better than environmental models. Depending on the algorithm used, they could explain between 17–73% of the variation in fungal phyla relative abundance and diversity (Fig. 3). The spectro-transfer functions of Glomeromycota produced $R^2$ values ranging from 0.17 using PLSR to 0.48 using 1D-CNN. The spectro-transfer functions of the Mortierellomycota and Mucoromycota produced the largest $R^2$ values ranging from 0.51 to 0.73 (Fig. 3).

Generally, PLSR and GPR were the least successful methods, while SVM, RF, Cubist and XGBoost were similarly successful for estimating fungal phyla relative abundance and diversity (Fig. 3). The 1D-CNN spectro-transfer functions were 13–31% more successful compared to other machine learning methods as they could explain between 45–73% of the variation in fungal relative abundance and diversity (Fig. 3).

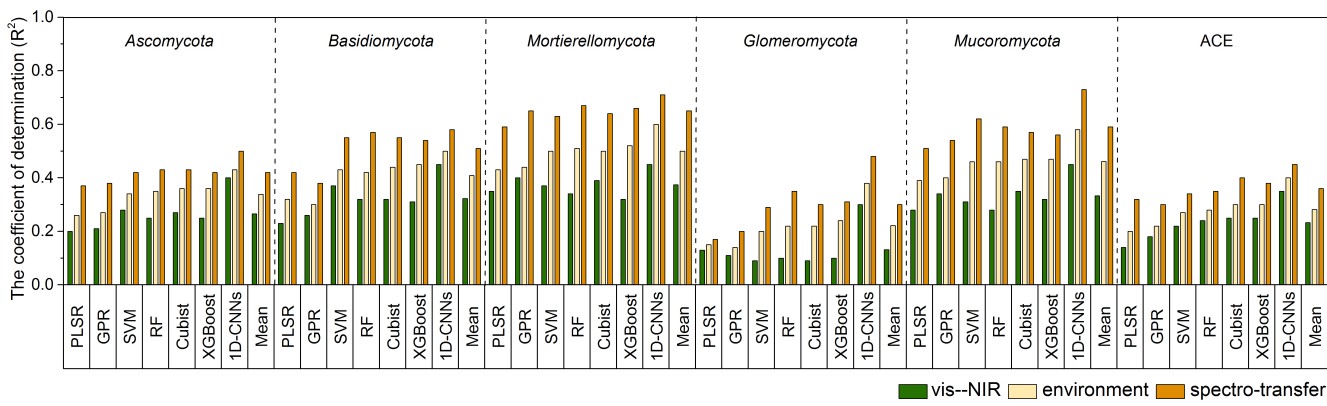

**Figure 3.** The coefficient of determination ($R^2$) for the vis–NIR spectroscopic models, soil and environmental models and the spectro-trasnfer functions that used combined set of the vis–NIR and readily available soil and environmental covariates, to estimate soil fungal phyla abundance and diversity (n = 577). The different statistical and machine learning methods were partial least squares regression (PLSR), gaussian process regression (GPR), support vector machines (SVM), random forest(RF), CUBIST, extreme gradient boost (XGBoost) and optimised 1D convolutional neural networks (1D-CNNs).

### 3.2    1D-CNNs spectro-transfer functions

The final architectures and optimised hyperparameters of the 1D-CNNs are given in Supplementary Table S3. As deep learning models are dataset dependent, the optimisation returned a different architecture for each response variable. Overall, the 1D-CNNs used simple architectures with less than 4 convolutional layers (Supplementary Table S3). Scatter plots of the measured versus estimated values of relative abundance and diversity using 1D-CNNs spectro-transfer functions and their validation statistics are shown in Fig. 4. Estimates of the relative abundance of Mortierellomycota and Mucoromycota produced $R^2$ values $\geq 0.60$, while estimates of Ascomycota and Basidiomycota produced $0.5 \leq R^2 < 0.6$. Estimates of Glomeromycota and ACE produced $0.4 \leq R^2 < 0.5$. The estimates were relatively unbiased (small ME), although generally small values were overestimated and large values were underestimated (Fig. 4). Imprecision contributed to the majority of the RMSE.

The imprecision of our estimates was a result of absence of repeated sampling and high adaptability of soil fungi to the wide range of environments.

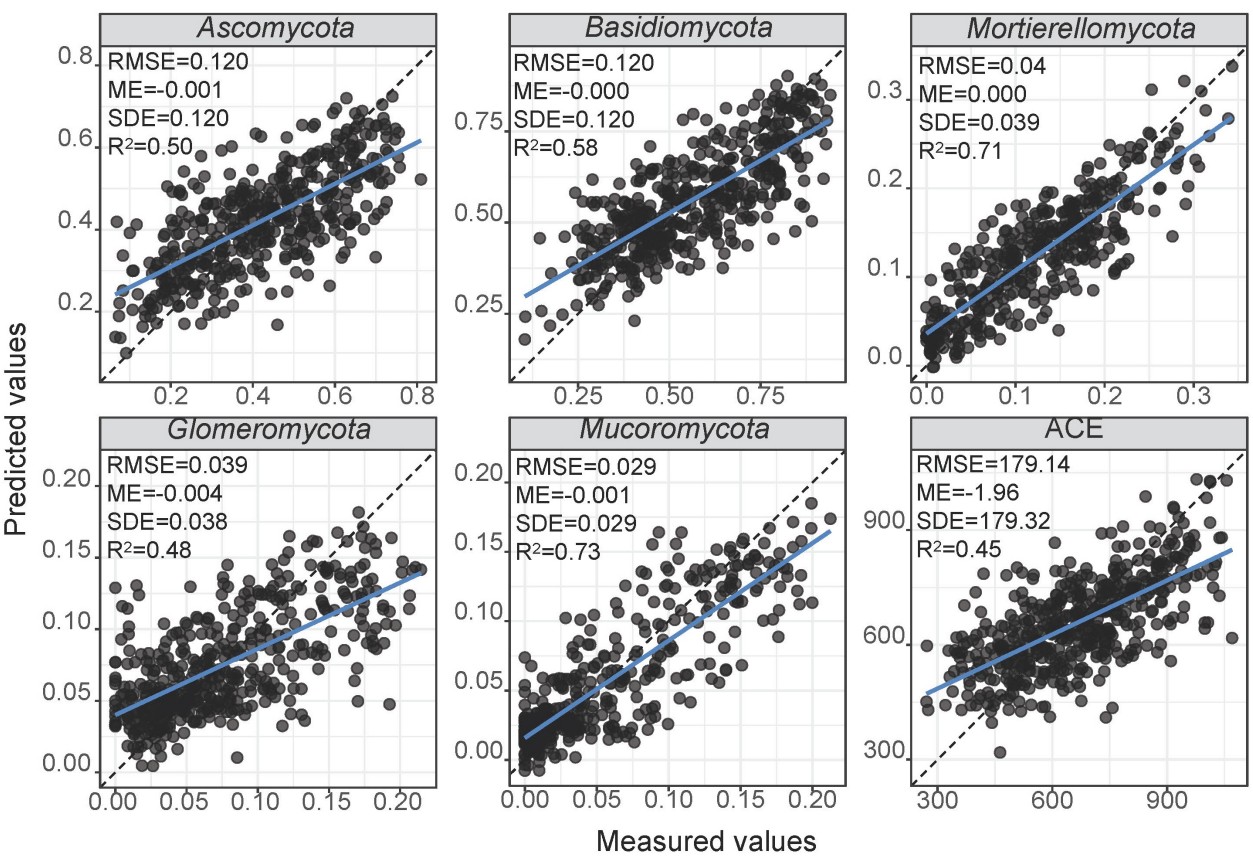

**Figure 4.** Performance of the CNN spectro-transfer functions for estimate of the relative abundance of dominant fungal phyla and diversity index. The spectro-transfer functions used vis–NIR spectra with other publicly available data on soil environmental variables. The plots show measured vs. estimated values using a 10-folds cross validation. The gray points represent no overlap with any other points, and the black points represent at least two points that overlap.

The important variables in the 1D-CNNs spectro-transfer functions of phyla relative abundance and diversity were vis–NIR wavelengths representing organic matter, iron oxide and clay minerals (Fig. 5).

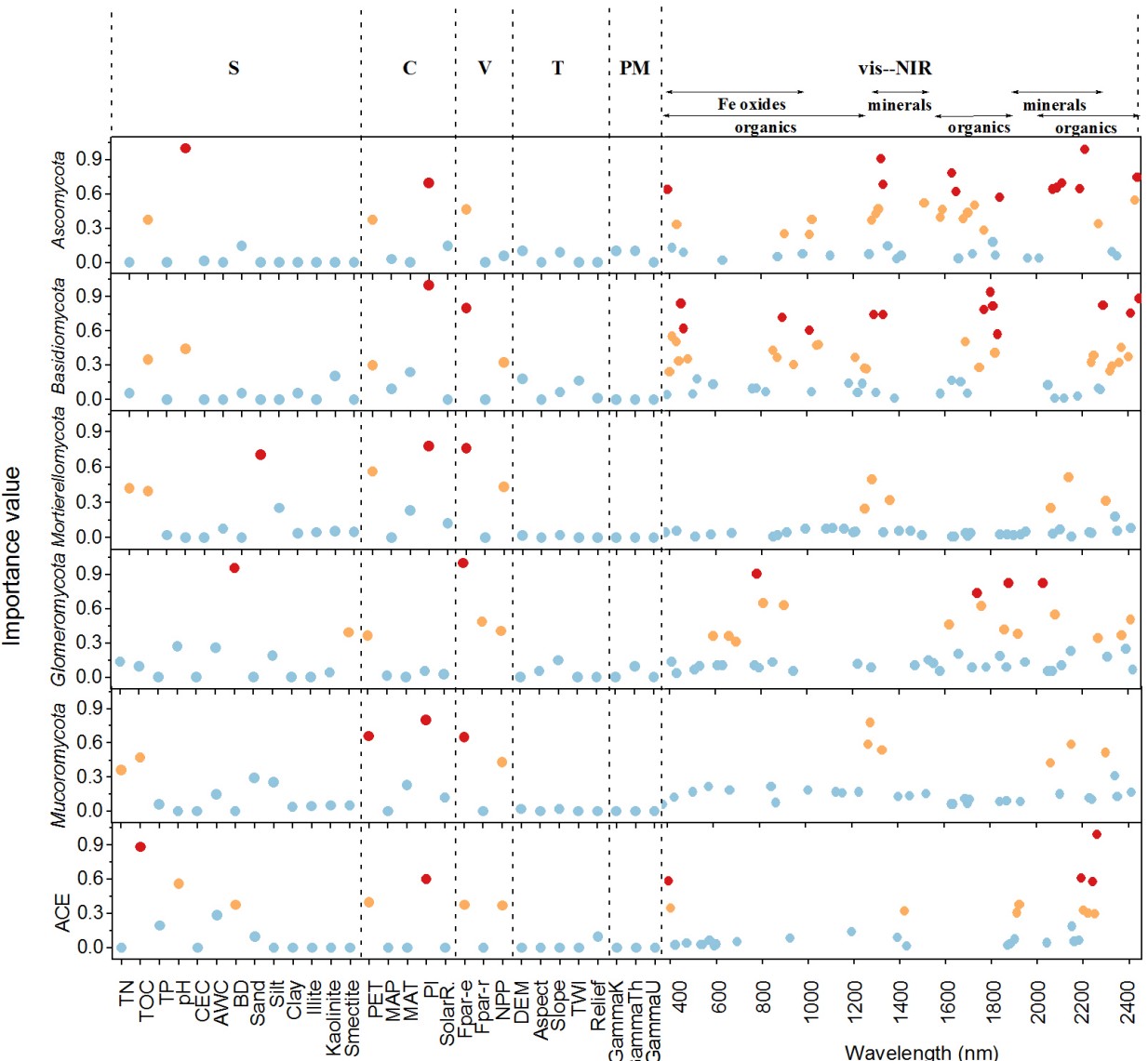

**Figure 5.** Important predictors of relative abundance of fungal phyla and diversity index measured by the variable importance of the 1D-CNNs spectro-transfer functions (n = 577) derived with publicly accessible data that represent soil (S), climate (C), vegetation (V), terrain (T), parent material (PM) and visible–near infrared (vis–NIR) spectra. The dots in red, orange, and blue color indicated the most, medium, and least important level. The importance value for the majority of wavelengths were low and close to zero value, thus these wavelengths were not shown to make the figure clearer.

The identified wavelengths mostly coincided with absorptions that are related to carbon functional groups found in organic matter, including C-H, N-H, C-O, with a smaller number of wavelengths coinciding with those that are related to clay minerals

and Fe-oxides (Table 2). The organic functional groups, C-H alkyl and methyls, N-H of amines, and C-O of carbohydrates, which might indicate the presence of relatively labile forms of carbon, and were important in the models of fungal phyla but not of ACE diversity. The C=O of amides and carboxylic acids, which represent stable forms of carbon were not as important in modeling (Fig. 5). Other wavelengths that represent Fe-oxides and clay minerals were also important in the models, indicating the different ecological niches and physiological characteristics (Table 2).

**Table 2.** Absorption band assignment for the most important vis–NIR wavelengths in the 1D-CNN models. The assignment of vis–NIR absorptions from Viscarra Rossel and Behrens (2010); Stenberg et al. (2010).

| | ACE | Ascomycota | Basidiomycota | Mortierellomycota | Glomeromycota | Mucoromycota |
|---|---|---|---|---|---|---|
| **Fe-oxides** | 390 | 390 | 410, 460 | | | |
| **Clay minerals** | 2190, 2240 | 1330, 2190, 2210 | 1330, 2140 | 1360, 2140 | | 1330, 2150 |
| **Organics** | | | | | | |
| C-H of aromatics | | 1630, 1650 | | | | |
| N-H of amine | | 2070, 2090, 2110 | 1010 | 2060 | 780, 2030 | 2060 |
| C-H of alkyl asymmetric-symmetric doublet | | | 890, 1290 | 1250, 1280 | 1740 | 1270, 1280 |
| C=O of carboxylic acids | | | | | | |
| C=O of amides | | | | | | |
| C-H of aliphatics | | | | | | |
| C-H of methyls | | 1840, 2440 | 1770, 1800, 1810 1830, 2450 | | 1880 | |
| C-OH of phenolics | | | | | | |
| C-O of carbohydrates | 2260 | | 2410, 2290 | 2300 | | 2300 |

Other soil properties, such as total organic carbon and pH were important variables in the spectro-transfer functions of Ascomycota and Basidiomycota, and fungal diversity. Total organic carbon and total nitrogen were important in the spectro-transfer functions of Mortierellomycota and Mucoromycota and bulk density was important in the spectro-transfer functions of Glomeromycota, Ascomycota and ACE diversity (Fig. 5). As well as soil properties, climatic factors such as the PI and PET, and vegetation, represented by Fpar-e and NPP were also important in the modelling of fungal phyla relative abundance and community diversity. The variables that we used to represent terrain, and parent material exerted less influence in the models (Fig. 5).

## 4 Discussion

Soil fungi play essential and diverse functional roles in ecosystem. However, they are challenging to investigate due to laborious, time-consuming and costly field sampling, and laboratory analysis. We show that spectro-transfer functions with readily accessible vis–NIR spectra and publicly available soil and environmental data could variably estimate ($R^2$ ranging from 0.45-0.73) soil fungal abundance and diversity measured with ITS gene metabarcoding. The spectro-transfer functions explained less than 60% of the variance in the two dominant phyla, the Ascomycota and Basidiomycota, representing 80% of the total fungal

relative abundance. In comparison, the spectro-transfer functions could explain more than 70% of the variance in the Mortierel-lomycota and Mucoromycota, which were less abundant in soil. The reason for the different predictability might be the coarse phylum-level identity. Compared with the Mortierellomycota and Mucoromycotawere, the Ascomycota and Basidiomycota are more complex phylogenetic classifications and consist of more diverse taxa with different phenotypic traits. These taxa have distinct ecological functions and environmental preferences, which might have reduced the predictability of their relative abundance at the phylum level. Classifying taxa with similar habitat preferences or studying at a finer taxonomic resolution might provide better predictability and understanding of soil fungal communities. The spectro-transfer function for the ACE index could only explain around 50% of the variance in diversity. The reason might be that local geography, environmental conditions, and difficult-to-proxy long-term natural selection and evolution affect community diversity.

The general concept of using proxies has been used in other studies to attempt more rapid estimation of microbial properties towards the diagnosis of soil quality. For example, Horrigue et al. (2016) developed a statistical predictive model of soil microbial biomass according to environmental parameters including soil physico-chemical and climatic characteristics across France. Their model ($R^2 = 0.67$) provided a reference value of microbial biomass for a given pedoclimatic condition to enable rapid diagnosis of soil quality across France. Other similar studies exist, for example Griffiths et al. (2016) who focused on the estimation of bacterial community structure and diversity at the Europe scale ITS gene metabarcoding analyses are expensive, laborious and require specialised laboratories and methods, while spectroscopic measurements are faster, less expensive, and soil-environmental data are more readily available. When many measures are needed, for example, to assess, characterise and improve our understanding of soil fungal communities and their associated functions at different scales, the approach could complement molecular techniques (Hart et al., 2020). For instance, to characterise spatial variation (i.e. for mapping), one needs many measurements that would be too expensive with only metabarcoding. In this case, estimates with the spectro-transfer functions ($R^2$=0.45–0.73) could complement the metabarcoding analysis to represent the variability present better. As a whole, the spatial characterisation will be more accurate than when only taking a few very precise measurements. This is the rationale for the characterisation of soil properties in space and time with sensing (Viscarra Rossel et al., 2011). The soil covariates in the model are derived from digital soil maps and not from measured soil samples. The reason is that using measured data would increase the cost of the approach significantly, making the approach less attractive. We note that the uncertainty in the spectro-transfer estimates caused by using the digital soil map predictors will propagate to the spectro-transfer functions and thereby lowering the precision of the estimates.

We do not expect that the spectro-transfer method will produce estimates that are as accurate as the more conventional molecular methods, even with further improvements in modelling and better covariates. This is because we understand that the modelling of living organisms is dynamic and hugely complex. Fungi vary over space and time (Duan et al., 2018), often showing that their prevalence in different habitats differs seasonally (Talley et al., 2002). The inconsistent correlations of fungi with climate and plant hosts observed in various ecosystems may be due to seasonal variation and spatial heterogeneity across single time point studies (Kivlin and Hawkes, 2016). Thus, temporal sampling is needed to capture the seasonal dynamics of microbial communities.

Our research uses soil fungal measurements at a single point in time and there are likely to be many undetermined controlling factors, including seasonal variability and complex biological interactions. Despite this drawback, our approach allows us to infer the distribution of soil fungal communities and diversity more simply and at a lesser cost, to help better understand the diversity and biogeography of soil fungi in different habitats. Thus, our approach shows promise and could complement molecular methods. We hope that our study will stimulate further research towards achieving more widespread characterisation of fungal abundance and diversity, which will help to deepen our understanding of fungal biology, biogeography and their environmental controls. Different spectra, new sensing technologies and improved methods could also improve the spectro-transfer approach.

Out of the seven statistical and machine learning models tested, the optimised 1D-CNNs were the most successful for estimating fungal phyla relative abundance and diversity, consistently producing the highest cross-validation $R^2$ values. The reason might be that the 1D-CNNs can automatically 'learn' the non-linear and complex relations between the soil fungal variables and the covariables. The models extract large features during convolution and adjust the weights of each covariate during the model iterations, which are also back-propagated (Breiman, 2001; Lecun et al., 2015). Although 1D-CNNs have been used for the spectroscopy modeling of soil physicochemical properties (Ng et al., 2019; Tsakiridis et al., 2020; Shen and Viscarra Rossel, 2021), to our best knowledge, this present study is the first to develop spectro-transfer functions for estimating soil fungal abundance and diversity.

Our results shown that the 1D-CNN spectroscopic models (with only vis–NIR spectra) could explain, on average, 40% of the variation in the relative abundance of fungal phyla and community diversity ($R^2$ values of 0.30–0.45). It is because these spectra characterise the soil's organic and mineral composition, which serves to supply energy and the elements that fungi use to promote vital activities (Müller, 2015). Microbial activities are closely associated with the types and amounts of organic matter and our results indicate that the most important vis–NIR wavelengths in the modelling of fungal relative abundance and community diversity corresponds to functional groups in the different types of organic compounds in the soils (Viscarra Rossel and Hicks, 2015) (Fig. 5 and Table S2 in Supplementary information).

The 1D-CNN spectro-transfer functions (with vis–NIR spectra and other soil and environmental data) improved the modelling. This suggests that other variables that represent climate, soil nutrients, pH, vegetation, are important predictors of fungal growth. Their use in the spectro-transfer functions provided additional and supplementary information for the modelling. On average, these models could explain 60% of the variation in abundance of fungal phyla relative abundance and diversity ($R^2$ values of 0.45–0.73).

The soil organic and mineral composition, represented by the vis–NIR spectra, were the most important predictors in the models for fungal relative abundance and community diversity. Additionally, total organic carbon and pH were important predictors of fungal diversity and the relative abundance of Ascomycota and Basidiomycota. Although most soil fungi do not require strict pH ranges for habitation and growth (Rousk et al., 2009; Zhao and Shen, 2018), some basophilic or acidophilic fungi are sensitive to changes in pH (Gai et al., 2006) and saprophytic fungi are thought to be more sensitive to soil pH, compared to other fungi (Kivlin and Hawkes, 2016). Soil bulk density was an important predictor of fungal diversity and the relative abundance of Glomeromycota. Many fungi, including those that form arbuscular mycorrhiza, such as Glomeromycota,

infect plants roots achieving mutualistic symbiosis (Schubler et al., 2001). Denser soil bulk density could reduce the availability of soil nutrients and water, leading to poor development of plant roots and a smaller infection rate for the symbiosis. The PI and evapotranspiration were the most important climatic predictors of fungal abundance and diversity in the models. PI represents the soil-water balance which has been shown to affect soil microbial growth at various studies (Bachar et al., 2010; Blankinship et al., 2011; Maestre et al., 2015; Delgadobaquerizo et al., 2018b). Because soil-water stress could strongly restrict microbial activity and distribution by controlling the availability of soil nutrients, pH and oxygen (Delgadobaquerizo et al., 2018b). NPP and Fpar-e were important predictors of fungal diversity and the relative abundance of the five dominant phyla. Larger values of NPP and Fpar-e occur due to greater biomass production and thus more accumulation of litter and coarse organic matter in soil. Soil fungi are some of the decomposers of litter and soil organic matter, including cellulose and lignin, which are often resistant to bacterial decomposition (Treseder and Lennon, 2015; Nicolas et al., 2019).

## 5   Conclusions

Our study contributes to the development of methods that could complement, not replace, molecular approaches for character-ising and better understanding the diversity and biogeography of soil fungi. We have shown that deep learning spectro-transfer functions are a promising new method for estimating soil fungal communities' relative abundance and diversity. The optimised 1D-CNNs outperformed the six other machine learning algorithms tested for estimating the relative abundance of fungal phyla and diversity. The spectro-transfer functions (with vis–NIR spectra and soil and environmental data) produced more accurate estimates ($R^2$ 0.45–0.73) than the spectroscopic models (only vis–NIR spectra; $R^2$ 0.36–0.55) and models with only the soil and environmental data ($R^2$ 0.38–0.60). As well as the soil organic and mineral composition, represented by vis–NIR spectra, other edaphic, climatic, and biotic factors including soil nutrients, pH, bulk density, potential evapotranspiration, the soil-water balance and net primary productivity were important predictors in the modelling. We hope that our study will provide food-for-thought for further research on the measurement and estimation of fungal abundance and diversity. We believe that improvements will be possible as new technologies and methodologies develop that will also help to deepen our understanding of fungal biology and biogeography.

*Code availability.*   The code used for the analyses presented in this work is available from the corresponding author on reasonable request.

*Data availability.*   The BASE data are available online at http://www.bioplatforms.com/soil-biodiversity/ after registration.

*Author contributions.*   R.A.V.R conceived the research and designed the study. Y.Y and Z.S. carried out the experiments. Y.Y and R.A.V.R. drafted the manuscript. R.A.V.R. edited the manuscript with input from all authors. A.B. derived the data on fungal relative abundance and diversity and edited the manuscript. All authors discussed and interpreted the results and produced the final manuscript.

*Competing interests.* The authors have no competing interests to declare.

*Acknowledgements.* This work was performed with funding from Curtin University. We acknowledge the contribution of the Biomes of Australian Soil Environments (BASE) consortium in the generation of data used in this publication. DOI 10.1186/s13742-016-0126-5. The 330 BASE project is supported by funding from Bioplatforms Australia through the Australian Government National Collaborative Research Infrastructure Strategy (NCRIS). The development of the convolutional neural network models was supported by resources provided by the Pawsey Supercomputing Centre with funding from the Australian Government and the Government of Western Australia.

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
