# Peer review of "Estimating soil fungal abundance and diversity at a macroecological scale with deep learning spectrotransfer functions"

_SOIL, 2021_

## Author Comment (AC1)

**Response to reviewers: Estimating soil fungal abundance and diversity at a macroecological scale with deep learning spectrotransfer functions by Yang et al.**

**Reviewer 1**

**General comments**

Trying to estimate soil fungi with vis-NIR spectroscopy is venturing down a very slippery slope. Soil fungal abundance and diversity have no spectrally active components. This is recognized by the authors in the Introduction. At best, estimation of soil fungal abundance and diversity is based on indirect correlation with properties that are spectrally active, such as clay minerals functional groups. However, it is not even clear from the literature whether soil fungal abundance and diversity have much relationship with these spectrally active soil components. With the use of complex machine learning models, there is a serious risk of finding accurate results simply because the model finds fortuitous relationships between spectra/variables and the property of interest. Without any surprise, in this study the model performing best is the most complex one: the deep learning model. The simplest model, PLSR, performed poorly in nearly all cases. PLSR is yet the benchmark method for spectroscopic modelling, and is not per se a simple model, it uses the whole spectrum and makes statistical decomposition of the spectral data (principal component analysis). This clearly suggests that the results of this study cannot be trusted because the good results obtained by the 1D-CNN models are not based on direct nor known indirect relationship with spectrally active soil components. At best, there is some kind of indirect relationship with spectrally active soil components which serve as proxy for the soil fungal abundance and diversity, but it is difficult to prove that this relationship indeed exists based on the literature. The reliability of this relationship would also then depend strongly on the relationship found among the data by the deep learning model and on the specific study case and

calibration data. For this reason alone, I do not endorse publication of this manuscript. There is no reason from the existing literature to justify the use of vis-NIR spectroscopy to estimate soil fungal abundance and diversity. It is misleading to claim that spectroscopy can "serve to supplement the more expensive and laborious molecular approaches".

**Authors:** Where to begin...let us see if we can help the reviewer get a grip as we would not like him/her to slip down the slope that he/she is on!

1. On the point that fungal abundance and diversity do not have 'spectrally active components'. Correct. Nowhere in our manuscript do we write that they do. Most functional soil properties, e.g. cation exchange capacity, pH, clay content, etc., do not have 'spectrally active components'. Yet, spectroscopy is used to estimate them because spectra hold information on clay minerals, iron oxides, carbonates, colour (and chromophores), organic functional groups present in organic matter, water, particle size. It is now well understood that estimates of such functinal properties are due to (first or higher-order) correlations to those 'spectrally active components'. It is in this manner that fungi can correlate to the spectra, albeit with rather poor predictability, as we show in Figure 3 (submitted ms). Hence, the idea for spetro-transfer functions, but more on this later.

2. Interestingly, the reviewer does not mention that the spectra measure organic functional groups in organic matter. Spectroscopy provides a chemical fingerprint of these functional groups (e.g., carbonyls, carboxyls, hydroxyls, phenols, and phenyls). Although there are no absorptions that can be directly assigned to soil fungi or microbial communities more generally, soil microbes are dependent on organic compounds for energy, but also on clay minerals and iron oxides for the supply of essential elements (Müller, 2015). For example, the decomposers of saprotrophic fungi are involved in decomposing complex organic material. They are dominant in soil containing more lignocellulose, recalcitrant

aromatic polymers that consist of phenylpropane units joined by C–C and ether linkages (Tunlid et al., 2013). Mutualists of mycorrhizal fungi are dominant in soil containing more decomposed lignin polymers and humic-rich organic matter that can form supramolecular aggregates. These aggregates are stabilised by hydrophobic interactions and hydrogen bonding (Tunlid et al., 2013). Thus, soil fungi are linked with 'spectrally active components' in soil, at the very least from the perspective of carbon supply.

3. The literature has shown that spectra can estimate biological soil properties (as described above). For example, various authors have shown that vis–NIR spectra can estimate microbial biomass, enzyme activity in soil, and respiration (Coûteaux et al., 2003; Cohen et al., 2005; Zornoza et al., 2008; Chodak, 2011). There have been several reports on estimating soil microbial community composition with vis–NIR reflectance spectroscopy too. For instance, Zornoza et al. (2008) and Davinic et al. (2012) performed vis–NIR modelling of groups of microorganisms measured with traditional PLFA profiling. They could well estimate ($R^2 > 0.7$) total PLFA biomass, total fungi, mycorrhizal fungi, total bacteria, Gram-positive bacteria and Actinomycetes. Using mid-infrared (mid-IR) spectroscopy, Nath et al. (2021) showed that they could estimate various soil biological properties. We found no research that uses vis–NIR spectra to infer soil microbial community composition based on 16S rRNA and ITS gene metabarcoding. Hence the added novelty of our study.

4. On the comments around PLSR and the machine learning methods. Glad to know the reviewer understands how PLSR and machine learning methods work. We do too, so we will keep this brief and refer to Figure 3 in the manuscript to make a few points that seem to have evaded the reviewer: (i) Our results clearly show that the spectra alone can not well estimate fungal phyla abundance and diversity, (ii) The soil and environmental data could better estimate fungi because they control the abundance and distribution of fungi—of course, this

has been well-reported (Delgadobaquerizo et al., 2018; Vetrovsky et al., 2019; Tedersoo et al., 2014 ), (iii) By combining the spectra with the soil-environment data (i.e. developing spectro-transfer functions), we could improve the modelling and achieve the best accuracies, and (iv) The PLSR estimates of the different phyla and ACE diversity were not too dissimilar to other machine learning methods, but yes, our optimised CNNs did perform somewhat better.

5. We have shown that spectro-transfer functions could fairly accurately (although as R2 notes, not superbly well) estimate fungal community composition and diversity measured with ITS gene metabarcoding using a large dataset from diverse ecosystems under different conditions. As stated in the manuscript, we are under no illusion that our method will replace metabarcoding. However, we propose that it could complement metabarcoding when many measurements are needed. ITS gene metabarcoding analyses are expensive, laborious and require specialised laboratories and methods. Once a spectro-transfer function is developed, vis–NIR spectroscopy is much easier, faster and less costly to measure, and soil-environmental data are more readily available. Since soil fungal properties are sensitive and reliable indicators of soil health, there has been much research to find variables related to soil fungal community to assess degradation processes, restoration strategies or management practices. Consequently, we think it worthwhile to research the development of spectro-transfer functions to model soil fungal composition and diversity.

6. Finally, we hope to have addressed the reviewer's concerns. We do not ask the reviewer to trust our models or us for that matter; however, we do ask that he/she reads our manuscript, tries to understand our results, interprets them accurately and provides us with a fair, unbiased assessment.

In addiction to this major conceptual problem, I have also several important comments that need to be addressed, please see below.

**Specific comments**

This paper is very similar to the Soil Biology and Biochemistry paper: same methodology, same data (fungi instead of bacteria, but still on the Australian BASE dataset), same covariates, same concept.

**Authors:** The general concept of the spectro-transfer functions is similar in both papers, but because we found modelling of fungi more challenging, we needed to research different methods and we thought useful to report them. This paper looks at fungal composition and diversity measured by ITS metabarcoding. To develop the spectro-transfer functions we tested different strategies (using spectra, environmental data, and a combination) and learning methods (7 different algorithms).

Combining spectral data and terrain into the modelling is not recommended and not common is spectroscopic research. The authors provide as input to the model a vector comprising both spectral data and environmental data found at site. The rational for doing this is unclear, and providing too much data to complex models will only aggravate the problem of finding spurious relationship among the data. This way of modelling brings additional problem with the covariates used. For example, the covariates called "mineralogy", i.e. kaolinite, illite and smectite (Appendix), are an interpolation of vis-NIR spectra band depth. So these covariates add redundant information. Combining spectral data and environmental variables into a single vector used for prediction is going too far into "unconscious" soil modelling.

**Authors:** We wonder who does not recommend using different data in modelling? and simply because it isn't commonly done, does the reviewer suggest that we shouldn't either? A couple of rather strange and subjective statements. Our rationale is clearly stated: we wished to test spectro-transfer functions. Of course, there are other examples in the soil and environmental sciences literature where spectra and other environmental data are combined in modelling. For example (amongst many other), Pullanagari et al. (2018) integrated airborne hyperspectral, topographic, and soil data for estimating pasture quality with Random Forest. Guo et al. (2019)

combined environmental factors and vis–NIR spectra to estimate soil organic matter with geospatial techniques. On the covariates, we see no issue with using data on soil clay minerals in the modelling, even if they were derived from specific absorptions in the near-infrared. The data were measured independently. We are unsure of the meaning of 'unconcious soil modelling', however, we hope that the reviewer is conscious that we perform the modelling using best-practice, and that we interpret the modelling (see Figure 5), finding that the interpretation was as reasonable as one might expect with this type of modelling.

To my understanding (because the writing needs to be improved and made more precise) the modelling is made on the absorbance spectra whereas the interpretation is made on a different model based on the continuum removed reflectance spectra. This is not correct. Interpretation should be made on the same model that the authors choose to be the best, not on a model fitted on different data. The authors are interpreting a model that was not used during modelling.

**Authors:** On the matter of the writing, it would serve the reviewer (and us) well if he/she could point out where and how our writing is lacking so that we might better address the comment. The manuscript was read and edited by the native English speaking authors as well as colleagues in our institutions before submission. We note that Reviewer 2 wrote that the paper is '...overall well written and is very well structured'. About the form of the spectra used for interpretation, there may be some misunderstanding here. Figure 2 is a stand-alone representation of the continuum removed spectra, which is simply to show that fungi actually show response in the vis–NIR. We used the continuum removed spectra because it helps to visualise the absorptions, more clearly than the absorbance first derivative spectra. Yes, the modelling was performed on the absorbance spectra, that is correct. Because the models were derived using absorbance spectra, their interpretation was made using these spectra and that interpretation is reported in Figure 5. To prevent confusion, we will include both types of spectra in Figure 2, the continuum removed and the

absorbance, first derivative spectra.

The BASE dataset contains more than 577 samples. Also, in the SBB paper 681 samples are used for bacteria. Why are not all data used? What is the reason to discard some observations? This should be clearly described in the Methods section of the manuscript.

**Authors:** We wrote this in the submitted manuscript, lines 86–90. When we calculated community diversity, to remove the bias induced by unbalanced sequencing, each sample was resampled at the same sequencing depth. The BASE dataset sought to produce as many sequences as resources allow with a minimum sequencing number of 10,000 per sample. Here, each sample was re-sampled at depth of 11,000 sequences to eliminate the unbalanced sequencing as shown in the Supplementary Figure 1. We chose 11,000 sequences as the resampling depth mainly because many samples only had this sequences number but also the rate of increase in the rarefaction curves is small at this depth. Doing this produced 577 data, which is therefore different that of the bacterial study.

The environmental variables used as covariates are outdated (Appendix). For example, the prescott index map for Australia is made at 5 km, but indeed downscaled at 90 m. The mineral maps (kaolinite, illite and smectite) are an interpolated product based on the band depths of vis-NIR spectra -should not be used here to avoid redundancy. The soil texture maps of 2015 are outdated, there are new ones since several months (see the paper https://www.publish.csiro.au/sr/pdf/SR20284). The vegetation maps are also very outdated. The authors used the old ones based on 250 m and 1 km resolution products, but there are new ones since several months based on Landsat 30 m data, see for example: http://data.auscover.org.au/xwiki/bin/view/Product+pages/Landsat+Seasonal+Fractional+Cover. Also, all the topographic covariates are now available at 30 m resolution and easy to download through the TERN repository: https://shiny.esoil.io/Covariates/.

**Authors:** The spatial data that we used is from around 2011–2017. The BASE soils were collected within that period so there is no reason to use newer data in this research. We see no reason why we should re-run all our analysis simply because there are newer maps of soil texture (Note that based on our assessments, when tested with independent data, the new texture maps provide insignificant improvements in overall accuracy compared to Figure 8 in Viscarra Rossel et al. (2015)). Similarly for the comments around resolution. Please note that the Prescott index layer that we used was produced at an approximate 90 m pixel resolution (by CSIRO terrain data analysts) using down-scaled climatic data. Also, why should we re-run all our analysis simply because there are newer, higher resolution datasets? We think it more important to use datasets of similar age. We have responded to the comments on the mineralogy maps.

The use of various machine learning methods is not of interest and there is the risk that this study simply becomes a comparison of validation statistics. Not clear what the added value is of applying seven methods. In this way, the manuscript runs the risk of being about the models and their comparison, rather than about understanding whether vis-NIR spectral data can effectively predict soil fungi. I think SOIL' readership is rather interested in the latter. Any model that here predicts better than another is case-study specific, and is unlikely to interest soil scientists who are the readers of this journal. Further, a model is usually chosen carefully with the problem in hand. Several of the models used by the authors are in fact very similar, they are all non-linear. The problem is here that none of the models described in the Appendix are thoroughly explained and it is not clear how the authors actually implemented all this. I could make many questions on each of the models, but my best advice here would be to reduce the model count to the minimum (PLSR and DL), otherwise most readers will probably see this manuscript as a programming exercise.

**Authors:** We disagree that the testing of the different methods is of no interest and

disagree with the suggestion to only report results for PLSR and DL. The purpose of our paper is to develop predictive spectro-transfer functions for fungal abundance and diversity and this requires us to test different algorithms for the modelling. Contrary to the comment that this might confuse readers about the purpose of the manuscript, our result show that all of the algorithms perform similarly (see Figure 3), and this should provide readers with confidence that the response captured by the modelling is 'real' and the modelling with the spectro-transfer functions robust. On the comment that 'a model is usually chosen carefully with the problem in hand'—this is not necessarily our experience with empirical modelling and machine learning. Usually, depending on the purpose, the datasets, etc., one needs to test different algorithms to find the most appropriate one. On the comment around the similarity of the algorithms, we tested one linear model (PLSR), three tree-based models (RF, XGBoost, Cubist) that work quite differently, SVM, Gaussian Processes and optimised CNNs. We chose those different methods because they provide a good and varied range of methods. We would be very surprised if after reading our manuscript readers thought that it was a 'programming exercise', particularly as we devote large portions of the results and discussion to interpreting our results and also provide Table 2 and Figure 5 for this purpose. We provide clear descriptions of the algorithms and their implementation in the Methods and Supplementary Information. Our descriptions are not exhaustive because these methods are not new and have been widely used and reported in other literature, which we cite.

It is unclear why the authors describe the accuracy statistics this way and what was done. With the information provided, the interpretation of these statistics is also unprecise. For example, the coefficient of determination of the linear model can be interpreted as amount of variance explained only in the case of a linear model with intercept. So the $R^2$ that the authors report in the manuscript is similar to a squared correlation coefficient, and should not be interpreted as percent of variance explained. See also the Wikipedia page, last paragraph of the intro:

`https://en.wikipedia.org/wiki/Coefficient$_$of$_$determination`. Further, decomposition of the RMSE into a bias and variance component is very well known in the literature and adds nothing to the paper. How the authors call it with "inaccuracy" and "imprecision" is not well accepted in the statistical literature. The decomposition is so well known that it is described in Wikipedia: `https://en.wikipedia.org/wiki/Mean$_$squared$_$error` or in any introductory statistic book, see, for example, page 298 in the book "Elementary Statistics for Geographer", by Burt et al. (2009), Third Edition. `https://www.routledge.com/Elementary-Statistics-for-Geographers/` `Burt-Barber-Rigby-Robeson-Horner/p/book/9781572304840`. Note that in both cases, they do not use terms such as "inaccuracy" and "imprecision" and I would agree not to use them.

**Authors:** Our reporting of the evaluation statistics are useful and correct. Reporting of the RMSE, ME and SDE describes the relevant errors because they represent the bias (ME), the imprecision (SDE) and thus the inaccuracy (RMSE) of the estimates, noting that $\text{RMSE}^2 = \text{ME}^2 + \text{SDE}^2$. That is, if the model predictions are unbiased, then imprecision will be the only contributor to their inaccuracy, and this seems to be the case for most of our results. We have cited Viscarra Rossel and Mcbratney (1998) who described the reporting of erros in this way, following the paper by Kempthorne and Allmaras (1965). The reviewer can find more detail in those texts. We additionally report the $R^2$, which is a more 'contextualized' measure of performance that can be used for more general assessments and comparisons to other work. $R^2$ we report definitely represent the proportion of the variation in the dependent variable that is predictable from the independent variable(s).

The methods section of the manuscript does not clearly describe what was done. For example, there are many methods available to compute the variable importance. At lines 137-140, the only description is "VarImp for caret" and permutation importance for the 1D-CNN models. This is not a description that allows one to

reproduce these results. What is done in caret VarImp? For the permutation importance of the 1D-CNN model, how is correlation among wavelengths accounted for (permutation is very sensitive to correlation)? How many permutations? What is the metric used (RMSE difference, ratio?)? What is the unit?

**Authors:** We disagree that our methods are unclear. The manuscript was also read by other colleagues before submission and Reviewer 2 could understand the methods, writing that the paper is '...overall well written and is very well structured'. However, we agree that the description of variable importance is a little brief. We will revise and improve the description of our implementation of the variable importance. Permutation importance measures the changes in a metric (e.g. RMSE) after a feature is permuted, where the relationship between the feature and the targets are broken. It can be applied on any estimator and does not require retraining the model. We used permutation importance and the RMSE) with 1000 permutations to calculate the variable importance of the CNN models. The importance values measures the RMSE increase after permutating the particular feature. In order to compare the importance between different fungal phyla and diversity, we scaled the importance values between 0 and 1. Note that we are aware of the issues with using permutation, however, we felt that for this study and with vis–NIR spectra, it would suffice if we used a large number of permutations to calculate the variable importance. We understand permutation importance has its limitations and that there are other methods to decipher CNNs, however they too have strengths and limitations (Fisher, et al., 2019). Although we think that it would be out of scope here, if the editor suggests, we could use another method that can better handle multicollinearity, e.g. SHAP (SHapley Additive exPlanations) (Lundberg and Lee, 2017). However, doing so might not necessarily improve our results or change our conclusions, but it would significantly complicate and lengthen the manuscript.

In Fig. 5, the importance should have a unit. What is the unit? How comes that covariates and spectra, which have very different units, can all have importance

values between 0 and 1? Usually a standardization can be made to the covariates and spectral data prior to use permutation on them. If this was done, please report it in the methods section and also use the same standardized input data for modelling. There is no reason to use different data and model for modelling and interpretation. **Authors:** The spectra and the covariates were centred and scaled before the modelling. This is clearly stated in the submitted manuscript line 126. For Permutation importance, please see our previous response.

**1 References**

Coûteaux, M.M., Berg, B., Rovira, P., 2003. Near infrared reflectance spectroscopy for determination of organic matter fractions including microbial biomass in coniferous forest soils. Soil Biology and Biochemistry, 35 (12), 1587–1600.

Cohen, M.J., Prenger, J.P., DeBusk, W.F., 2005. Visible-near infrared reflectance spectroscopy for rapid, nondestructive assessment of wetland soil quality. Journal of Environmental Quality, 34(4), 1422–1434.

Chodak, M., 2011. Near-infrared spectroscopy for rapid estimation of microbial properties in reclaimed mine soils. Journal of Plant Nutrition and Soil Science, 174 (5), 702–709.

Davinic, M., Fultz, L.M., Acosta-Martinez, V., et al., 2012. Pyrosequencing and mid-infrared spectroscopy reveal distinct aggregate stratification of soil bacterial communities and organic matter composition. Soil Biology and Biochemistry, 46 (1), 63–72.

Delgadobaquerizo, M., Reith, F., Dennis, P. G., et al., 2018. Ecological drivers of soil microbial diversity and soil biological networks in the Southern Hemisphere, Ecology, 99, 583–596.

Fisher, A., Rudin, C. and Dominici, F., 2019. All Models are Wrong, but Many are Useful: Learning a Variable's Importance by Studying an Entire Class of Prediction Models Simultaneously. J. Mach. Learn. Res., 20(177), pp.1-81

Guo, L., Zhang, H., Chen, Y. et al.,2019. Combining Environmental Factors and Lab VNIR Spectral Data to Predict SOM by Geospatial Techniques. Chin. Geogr. Sci., 29, 258–269.

Horwitz B., Mukherjee P., Mukherjee M., Kubicek C. (eds) Genomics of Soil- and Plant-Associated Fungi. Soil Biology, vol 36. Springer, Berlin, Heidelberg.

Kempthorne, O., Allmaras, R. R., 1965. Errors of observation. In 'Methods of Soil Analysis. Part 1. Physical and Mineralogical Properties, Including Statistics of Measurement and Sampling'. Agronomy Monograph No. 9.(Ed. C. A. Black.) (ASA/SSSA: Madison, WI, USA.)

Lundberg, Scott M., and Su-In Lee. 2017. A unified approach to interpreting model predictions. Advances in Neural Information Processing Systems.

Müller, B., 2015. Experimental interactions between clay minerals and bacteria: a review, Pedosphere, 25, 799–810.

Nath, D., Laik,R., Meena,V. S., et al., 2021. Can mid-infrared (mid-IR) spectroscopy evaluate soil conditions by predicting soil biological properties? Soil Security, 4, 100008.

Pullanagari, R., Kereszturi,G., Yule, I. 2018. Integrating Airborne Hyperspectral, Topographic, and Soil Data for Estimating Pasture Quality Using Recursive Feature Elimination with Random Forest Regression" Remote Sensing, 10(7), 1117.

Tedersoo, L., Bahram, M., Polme, S., et al., 2014. Fungal biogeography. Global diversity and geography of soil fungi. Science, 346 (6213), 1256688.

Tunlid A. et al., 2013. Genomics and Spectroscopy Provide Novel Insights into the Mechanisms of Litter Decomposition and Nitrogen Assimilation by Ectomycorrhizal Fungi. In:

Vetrovsky, T., Kohout, P., Kopecky, M., et al., 2019. A meta-analysis of global fungal distribution reveals climate-driven patterns., Nature Communications, 10, 5142.

Viscarra Rossel, R. A. and McBratney, A. B., 1998.Soil chemical analytical accuracy and costs: implications from precision agriculture. Australian Journal of Experimental Agriculture, 38, 765–775.

Zornoza, R., Guerrero, C., Mataix-Solera, J., et al., 2008. Near infrared spectroscopy for determination of various physical, chemical and biochemical properties in mediterranean soils. Soil Biology and Biochemistry. 40 (7), 1923–1930.

---

## Author Comment (AC2)

**Response to reviewers: Estimating soil fungal abundance and diversity at a macroecological scale with deep learning spectrotransfer functions by Yang et al.**

**Reviewer 2**

**General comments**

The manuscript by Yang and colleagues describes a study whose aim is to test the predictive capabilities of learning models based on different types of predictors such as near-infrared spectroscopy (point data) and large-scale environmental variables with respect to the abundance and diversity of soil fungi.

The study is based on a rather large Australian dataset, comprising several hundred soil samples from different regions of Australia under different land-use types, for which fungal abundance and diversity have already been measured and published in a previous study, and for which near infrared (NIR) spectra (and related environmental data) have also been acquired and published in previous papers.

The authors test different strategies to build their learning models, varying the type of predictors used (spectroscopic data alone, environmental data alone, or a combination of both types of information) and the learning algorithms used (7 different algorithms). The performances of the different learning models to predict the abundance and diversity of soil fungi are tested by cross-validation.

The objective of this manuscript is of great interest to SOIL journal readers: are we able to predict the abundance and diversity of soil fungi by learning models based on simpler-to-obtain predictors in unknown Australian soils?

The article is overall well written and is very well structured. The figures and tables are clear. The learning models seem to have been well constructed by a team that is very well versed in data science techniques and the application of spectroscopy technique to soils.

**Authors:** We thank the reviewer for reading our manuscript and making an effort to

understand our work. We share the reviewers view that our work is of interest to the readership of SOIL and we are grateful for the overall positive comments.

**Specific comments**

However, in reading this article, several major shortcomings appeared to me.

Seasonality of the abundance and diversity of soil organisms First, an important point: it seems to me that the seasonal variability of the abundance and diversity of soil organisms in general and soil fungi in particular is a major "unthought" of this manuscript. In this work, no reference to this major determinant of fungal abundance and diversity and no information on the date of sampling and climatic conditions in the month prior to sampling of each soil sample was collected and exploited as a potential predictor by the authors in their learning models. It seems to me that through this "unthought" of seasonallity, the authors give up, without ever informing the reader, an important part of the determinism of their soil fungus-related variables of interest (cf. e.g. reference Kivlin and Hawkes, 2016 which is cited by the authors in their manuscript). In this paper we can read: "The strong seasonal pattern in fungal richness and abundance suggests that fungal studies in tropical forests require temporal sampling to capture the full community. Indeed, the inconsistent correlations of fungi with climate and plant hosts across tropical ecosystems [...] may be due to seasonal variation and spatial heterogeneity across single time point studies" (Kivlin and Hawkes, 2016). In general, how can a soil biodiversity measurement at a time t of a soil can reliably represent the diversity and relative abundance of different species/groups of soil fungi for that soil in its different seasons? Applied to the authors' objective in this manuscript: how can one hope to reliably predict very dynamic soil variables (fungal organism abundance and diversity) with predictor variables that are much less dynamic in the soil or in the environment? At best, one could hope to predict large differences between very contrasting soil pedoclimates: something that Biogeography of soil organisms works have often already identified. However, such Biogeographic works do not constitute

predictive models of soil biodiversity at a given time t. At the very least, it seems to me that this point, which is an important intrinsic limitation of this study (and also of a similar previous study focusing on soil bacteria by Yang et al. 2019 in Soil Biology and Biochemistry; `https://doi.org/10.1016/j.soilbio.2018.11.005`; should be discussed in detail by the authors.

**Authors:** We thank the reviewer for the comment and suggestion. Of course, we agree with the general comment on the seasonal variability of microorganisms. We will highlight in a revised discussion the shortcomings of any modelling when not accounting for seasonality of the abundance and diversity of soil fungi, also with reference to Kivlin and Hawkes (2016). To make the point clear, we will revise the introduction leading to our aims with something like: '...Fungal abundance and diversity are dynamic in space and time (Duan et al., 2018), often showing that habitat prevalence differs seasonally (Talley et al., 2002).The inconsistent correlations of fungi with climate and plant hosts observed in various ecosystems may be due to seasonal variation and spatial heterogeneity across single time point studies (Kivlin and Hawkes, 2016). Thus, temporal sampling and seasonal climate data are needed to capture the seasonal dynamics of microbial communities. In our study, the development of spectro-transfer functions with vis–NIR spectra and machine learning is based on soil fungal measurements at a single time. Despite this limitation, our approach provides a possible opportunity to infer the continuous distribution of soil fungal communities and diversity more simply and at a smaller cost. Thus it can help to better understand the diversity and biogeography of soil fungi in different habitats.....'. Additionally, for completeness, we could also add information on the climate at the time of sampling and in the month prior to sampling.

Overly optimistic conclusions compared to reported results A second major point concerns the conclusions that the authors draw from their results regarding the predictive capabilities of the learning models they have constructed for the abundance and diversity of soil fungal groups in Australia. These conclusions seem to

me to be overly optimistic compared to the results the authors show in their manuscript. After reading this paper, if I had to answer the question: can we, in unknown Australian soil samples, robustly estimate the abundance and diversity of major soil fungal groups with the learning models constructed in this work, my answer would be: no, and we're a long way off. At most we can identify some major soil and environmental determinants of the abundance of these groups and their diversity, making this a work of Biogeography of soil organisms, much more than a paper that aims to predict these properties in unknown soils to study soil health.

Indeed, in the introduction to their article, the authors mention that "a paucity in the availability of soil microbial data is thought to be one of the main contributors to the uncertainty of soil health assessment and ecosystem management" while in concluding their paper, the authors state that "here, we show that spectro-transfer functions with readily accessible vis–NIR spectra and publicly available soil and environmental data can be developed to estimate soil fungal abundance and diversity" I disagree with the conclusions of the authors: using their models for soil health assessment of a particular Australian soil (for a criteria of soil health linked to soil fungal abundance or diversity for instance) would certainly not help improving the acuracy of this assessment given the predictive performance showed in this manuscript.

Specifically, the models with the best predictive capabilities use a combination of point spectroscopic data and continuous environmental data, and they most often use the deep learning algorithm 1D-CNNs. But despite these (still somewhat obscure) performance differences between the different algorithms, even the "best" results shown by Yang and colleagues are not very encouraging in terms of the actual predictive capabilities of the learning models built on unknown Australian soil samples:

- For the two (very) dominant soil fungus groups in Australia (Ascomycota; Basidiomycota) representing on average 80% of the total fungal abundance; Table 1),

the "best" learning models show $R^2$ below 0.6. I infer that the learning models constructed in this paper do not robustly quantify the abundance of groups representing 80% of the fungi in Australian soils.

- For the diversity of fungi estimated by the "ACE" indicator, the $R^2$is 0.45, I deduce that the learning models built in this article do not allow to quantify in a robust way the diversity of fungi in Australian soils.

- A few models with slightly better predictive abilities are shown for the abundance of two groups of very low abundance fungi in soils (although it is difficult to quickly judge their performance in the absence of synthetic performance indicators such as the ratio of performance to deviation or the ratio of performance to interquartile distance).

**Authors:** We agree with the reviewer's concern. Generally, we are some way off very accurate estimates of fungal abundance and diversity with our method. We understand that the large unexplained variance of fungal abundance and diversity were mainly because of the complexity of the living organism itself and large undetermined controlling factors, including seasonal variability (see previous comments and responses). We will make it clear in the revision that in this study we presents a promising method to estimate fungal communities and diversity. We hope that our study will provide food for thought and guide further research towards achieving more robust fungal abundance and diversity predictions, which we think will be possible with our deepening understanding of fungal biology, biogeography, the improved understanding of controlling factors and with the development of new technologies and methods. Nevertheless, the spectro-transfer functions we constructed could fairly accurately (as per the reported statistics) estimate fungal community composition and diversity measured with ITS gene metabarcoding. As we stated in the manuscript, we do not think that our method will replace metabarcoding, however it could complement it when many measurements are needed. Our hypothesis here is that even if some of our spectro-transfer functions are

relatively innacurate (as per our Figure 4), when one has many samples to measure across an area, the estimates will better represent the variability present and as a whole be more accurate than when only taking a few very accurate measurements. We can strengthen the discussion around this point, however, testing of this hypothesis will need to be for a future study. ITS gene metabarcoding analyses are expensive, laborious and require specialised laboratories and methods, while vis–NIR spectroscopy is relatively faster and less expensive, and soil-environmental data are more readily available. Using soil sensing technologies, such as spectroscopy together with molecular approaches could greatly improve the utility of microbial inventory data (Hart et al., 2020). For example, Delgadobaquerizo et al.(2018) mapped the global distribution of the soil bacterial composition based on only 237 bacteria inventory data in the world. The global spectroscopy data is very abundant and reach tens of thousands. With the use of spectro-transfer functions, the bacterial data density will be improved, then benefits the more accurate digital soil mapping of bacteria or fungi. Thus, we will tone down and clarify our conclusions on the predictive capabilities of the learning models constructed.

Other points

Section 2.1: ecosystem types, please clarify the difference between woodland and forest.

**Authors:** Thank you, we will make the distinction. The term woodland is generally used in Australia to describe ecosystems which contain widely spaced trees, the crowns of which do not touch. Woodland consists of areas with fewer and more scattered trees than forests. In temperate Australia, woodlands are mainly dominated by Eucalyptus species. Temperate woodlands occur predominantly in regions with a mean annual rainfall of between 250–800mm, forming a transitional zone between the higher rainfall forested margins of the continent and the shrub and grasslands of the arid interior.

Section 3.1:   In total, more than 60 million quality filtered sequences in the

whole dataset were obtained, with an average of 107 310 sequences per sample. When we clustered the sequences at 97% similarity level 202200 OTUs were detected. Each sample had an average of 666 OTUs : this was already presented in the section 2.2, please remove it from the results section.

**Authors:** We thank the reviewer for picking this up. We will revise it.

Figure 1b: please add some information to remind reader that this graph shows mean abundances (this graph does not represent errors on the mean, which is huged as shown in Table 1).

**Authors:** Thanks you, agree and we will revise .

I agree with R1 that interpreting NIR spectra with continuum removed reflectance signals and using savitsky-golay first derivatives of absorbance signals in the modelling work can be misleading. Please interpret the NIR spectra with the signal used for modelling.

**Authors:** We refer the reviewer to our response to R1 on this matter. We will add the SG1Der spectra too.

I wonder how does the sum of the model predictions for the relative abundance of the 5 main groups of soil fungi behave for the soil sample set used in this study: does this sum get close to 1 for all soil samples?

**Authors:** Thank you for the question. No, the sum of the model predictions for the relative abundance of the 5 main groups of soil fungi were not close to 1 for all soil samples. There are samples with the sum larger than 1. Our models did nothing to constrain the cumulative abundance.

**References**

Delgado-Baquerizo, M., Oliverio, A. M., Brewer, T. E., et al, 2018. A global atlas of the dominant bacteria found in soil. Science, 359 (6373), 320-325.

Duan, Y, Xie N, Song, Z,et al., 2018. A High-Resolution Time Series Reveals Distinct Seasonal Patterns of Planktonic Fungi at a Temperate Coastal Ocean Site (Beaufort,

North Carolina, USA). Appl Environ Microbiol., 84(21), e00967-18.

Hart, M. M., Cross, A. T., D'Agui, H. M.,et al., 2020.Examining assumptions of soil microbial ecology in the monitoring of ecological restoration, Ecological Solutions and Evidence, 1, e12031.

Kivlin, S. N., Hawkes, C. V., 2016. Tree species, spatial heterogeneity, and seasonality drive soil fungal abundance, richness, and composition in Neotropical rainforests, Environmental Microbiology, 18, 4662–4673, 201.

Talley, S.M., Coley, P.D., Kursar, T.A. 2002. The effects of weather on fungal abundance and richness among 25 communities in the Intermountain West. BMC Ecol., 2, 7.

---

## Author Response (AR1)

**Response to reviewers: Estimating soil fungal abundance and diversity at a macroecological scale with deep learning spectrotransfer functions by Yang et al.**

We thank the topical editor and reviewers for their comments. Below we provide detailed responses (in blue text and preceded by **Authors:**), indicating the changes we made in our revision. First, we address the topical editor's comments and then the reviewers.

**Topical Editor: Comments to the author**

Two reviewers have evaluated the manuscript and suggested several points to improve the manuscript. Thank you for taking up most of these points in your reply and a revised version. Please not that a revised version is generally only due after the paper discussion period and the editorial board decision on the paper.

In the comment to R1 already posted on the site, it seems to me that you have not written clearly what changes you intend to make.

**Authors:** The changes that we proposed relate to the representation of the spectra in Figure 2, the R$^2$ used and an improved explanation of the variable importance method. Please see below for details.

I beleive R1 is advocating for parsimonious and understandable models. It is generally advised to run a covariates selection step before interpretation of the models see for example https://doi.org/10.5194/soil-7-217-2021.

**Authors:** Thank you for the comment. We agree that it is generally sensible to perform a variable selection prior to modelling, particularly when there are hundreds of geographical predictors and when employing the environmental correlation approach in spatial machine learning (like in the example provided by the editor).

Spectroscopic modelling is somewhat different and variable selection tends not to work all that well when developing predictive models—hence the preference for dimensionality reduction methods for multivariate calibrations, such as partial least squares regression (PLSR). Our experience with machine learning is similar in that variable selection prior to modelling tends to produce models that explain less variance that models that use all of the wavelengths. It might be the 'non-specificity' and collinearity of the wavelengths for modelling soil properties. When we were developing the research and during our initial analyses, we did perform a variable selection using different methods to select important wavelengths and the Boruta algorithm performed best. However, we found that the models did not perform as well as when we used all wavelengths. Table 1 below, shows these results for PLSR and CNNs, and shows that models that used all of the wavelengths (full-spectrum+DSM+ENV) accounted for more variance in fungal phyla abundance and diversities than the models with only selected variables (spectrum+DSM+ENV). Please note that we have result for all of the algorithms tested, however, to illustrate our response, we show results only for PLSR and for the more complex CNN approach. Therefore, based on these results, in the manuscript we report only the full spectrum results.

Table 1: Comparison of goodness of estimates by a 10-fold cross validation based on PLSR and 1D-CNNs.

| Variables | PLSR | | 1D-CNNs | |
|---|---|---|---|---|
| | Full-spectrum + DSM + ENV ($R^2$) | Boruta selection + DSM + ENV ($R^2$) | Full-spectrum + DSM + ENV ($R^2$) | Boruta selection + DSM + ENV ($R^2$) |
| *Abundance* | | | | |
| *Ascomycota* | 0.37 | 0.33 | 0.50 | 0.45 |
| *Basidiomycota* | 0.42 | 0.39 | 0.58 | 0.52 |
| *Mortierellomycota* | 0.59 | 0.53 | 0.71 | 0.65 |
| *Glomeromycota* | 0.17 | 0.15 | 0.48 | 0.36 |
| *Mucoromycota* | 0.51 | 0.45 | 0.73 | 0.62 |
| *Diversity* | | | | |
| ACE | 0.32 | 0.30 | 0.45 | 0.41 |

Generally, the two reviewers suggested more balanced analysis and interpretation. I think that both readers agree that the conclusions are too much optimistic.

**Authors:** We understand Reviewer 2's comment about the somewhat 'optimistic' conclusions. We have clarified and toned down our conclusions on the predictive capabilities of the spectro-transfer functions. We revised the relevant sections of the discussion as follows:

- We revised the discussion around the estimation with the spectro-transfer functions: "...We show that spectro-transfer functions with readily accessible vis–NIR spectra and publicly available soil and environmental data could variably estimate (with $R^2$ ranging from 0.45–0.73) soil fungal abundance and diversity measured with ITS gene metabarcoding...". Note that here we report the $R^2$ range without subjective assessments on the predictability of the models.

- We clarified how these models with varying predictability could complement (not replace) molecular approaches for the assessment, characterization and improved understanding of soil fungal communities. See below, in response to R2.

- We make it clear that our method and results are **encouraging** (not a fait accompli) with potential to help characterise fungal communities and diversity (when used together with the more difficult-to-measure molecular methods). See below, in response to R2.

**Authors:** To provide a more balanced analysis of our results, we also cite other studies that use the general concept of proxies to achieve rapid estimates of other microbial properties to help diagnosis of soil quality. See below, in response to R2.

Several aspects need some further attention: · the use of a more synthetic performance indicators instead of the R2. I think it is important to give the formula

of the R2 you used as it is possible to confuse between 2 formulas. The model efficiency coefficient MEC (Janssen and Heuberger, 1995) is equal to the fraction of the explained variance based on the 1:1 line of predicted versus observed that is defined as 1 minus the ratio between residual sum of squares and total sum of squares. Did you use this one ?

**Authors:** Yes, we used the Sutcliffe model efficiency and have specified this in the methods. Please see below, in response to R1.

· The use of the soil covariates in the model is derived from the 90m DSM products not from the soil samples, therefore it may decrease the robustness of the pedo-transfer function. Please, could you add a discussion on this aspect.

**Authors:** We agree that using the estimates from the digital soil mapping products will not be as good as using measured data. However, the idea of the spectro-transfer functions (much like the more conventional PTFs) is to use data that is inexpensive/free and readily available. Using measured data would increase the cost of the approach significantly and in that case, it might make more sense to use the molecular methods directly. We take the general point though, so we added some discussion as follows: "... The soil covariates in the model are derived from digital soil maps and not from measured soil samples. The reason is that using measured data would increase the cost of the approach significantly, making the approach less attractive. We note that the uncertainty in the spectro-transfer estimates caused by using the digital soil map predictors will propagate to the spectro-transfer functions and thereby lowering the precision of the estimates..."

· I am less concerned than R1 about the overlap with previous paper as the modelling of fungi may raise other issues than modelling of bacteria. It is genarally admited that Fungi are for example not as dependent on specific plant species as some bacteria (https://soilquality.org.au/factsheets/soil-bacteria-and-fungi-nsw).

**Authors:** Thank you, that is also our position.

We are looking forward to a revised version of your manuscript. Yours sincerely, Nicolas

**Revisions based on reviewers comments**

Below, we detail the revisions made as per our responses to each reviewer's comments in the discussion.

**Revisions based on reviewer1's comments**

**Authors:**

- To prevent confusion between visualization and interpretation of spectra, we included both types of spectra in Figure 2, the continuum removed and the absorbance, first derivative spectra. We revised the relevant sections in the Methods and Results.

- We clarified in manuscript that use used the Nash Sutcliffe model efficiency $R^2$ as follows: "...We evaluated the estimates using the Nash Sutcliffe model efficiency, other wise known as the coefficient of determination ($R^2$), which represent the fraction of the explained variance based on the 1:1 line of estimated versus measured values(Janssen and Heuberger, 1995). The $R^2$ was computed as 1-RSS/TSS, where RSS is the residual sum of squares and TSS is the total sum of squares."

- We revised and improved the description of our implementation of the variable importance as follows: "...To calculate the variable importance of the CNN models, we used permutation variable importance. In our case, we run 1000 permutations and measured the decrease in RMSE after a predictor was permuted (randomly rearranged). The permutation breaks the relationship

between the predictor and the response variables, and a reduction in RMSE indicates how much the model depends on the particular predictor. An advantage of this approach is that it can be applied on any estimator and does not require retraining the model (Breiman, 2001; Fisher et al., 2019). In order to compare the importance between different fungal phyla and diversity, we scaled the importance values between 0 and 1."

**Revisions based on reviewer2's comments**

**Authors:** In summary, Reviewer 2 has two main concerns with our work: (i) the lack of a discussion on the seasonal variability of microorganisms and (ii) his perceived 'over-optimistic' predictability of the spectro-trasnfer functions. He also has some other points for us to consider. Below, we address each of the comments and suggestions made.

**Authors:** Addressing the comments on the seasonal variability of microorganisms :

- To clarify, we added discussion on the seasonal variability as follows: "...Fungi vary over space and time (Duan et al., 2018), often showing that their prevalence in different habitats differs seasonally (Talley et al., 2002). The inconsistent correlations of fungi with climate and plant hosts observed in various ecosystems may be due to seasonal variation and spatial heterogeneity across single time point studies (Kivlin and Hawkes, 2016). Thus, temporal sampling is needed to capture the seasonal dynamics of microbial communities. Our research uses soil fungal measurements at a single point in time. Despite this drawback, our approach allows us to infer the distribution of soil fungal communities and diversity more simply and at a lesser cost, to help better understand the diversity and biogeography of soil fungi in different habitats..."

- In the Methods, we now include general information on seasonality and climate at the time of sampling, as follows: "...In that project, sampling were

undertaken from soil that supports diverse plant communities across Australia. The sampling was carried out during the growing season when hydrothermal conditions are most conducive to typical plant growth. In the higher rainfall forested regions of the continent, the soil samples were collected mostly in spring and summer from September to February. In the shrublands and grasslands of the semi-arid and arid interior, soil samples were collected in spring from September to November. In the transitional zone between the southeast coast and the more arid interior, soil samples were collected in mainly autumn from March to May..."

**Authors:** To address the comment on the 'over-optimistic' predictability of the spectro-transfer functions, we have toned down and clarified our discussion as follows:

- "...Soil fungi play essential and diverse functional roles in ecosystem. However, they are challenging to investigate due to laborious, time-consuming and costly field sampling, and laboratory analysis. We show that spectro-transfer functions with readily accessible vis–NIR spectra and publicly available soil and environmental data could variably estimate (with $R^2$ ranging from 0.45-0.73) soil fungal abundance and diversity measured with ITS gene metabarcoding. The general concept of using proxies has been used in other studies to attempt more rapid estimation of microbial properties towards the diagnosis of soil quality. For example, Horrigue et al.(2016) developed a statistical predictive model of soil microbial biomass according to environmental parameters including soil physico-chemical and climatic characteristics across France. Their model ($R^2 = 0.67$) provided a reference value of microbial biomass for a given pedoclimatic condition to enable rapid diagnosis of soil quality across France. Other similar studies exist, for example Griffiths et al. (2016) who focused on the estimation of bacterial community structure and diversity at the Europe scale..."

- "...ITS gene metabarcoding analyses are expensive, laborious and require

specialised laboratories and methods, while spectroscopic measurements are faster, less expensive, and soil-environmental data are more readily available. When many measures are needed, for example, to assess, characterise and improve our understanding of soil fungal communities and their associated functions at different scales, the approach could complement molecular techniques (Hart et al., 2020). For instance, to characterise spatial variation (i.e. for mapping), one needs many measurements that would be too expensive with only metabarcoding. In this case, estimates with the spectro-transfer functions ($R^2$=0.45–0.73) could complement the metabarcoding analysis to represent the variability present better. As a whole, the spatial characterisation will be more accurate than when only taking a few very precise measurements. This is the rationale for the characterisation of soil properties in space and time with sensing (Viscarra Rossel et al., 2011)."

- "...We do not expect that the spectro-transfer method will produce estimates that are as accurate as the more conventional molecular methods, even with further improvements in modelling and better covariates. This is because we understand that the modelling of living organisms is dynamic and hugely complex. Fungi vary over space and time (Duan et al., 2018), often showing that their prevalence in different habitats differs seasonally (Talley et al., 2002). The inconsistent correlations of fungi with climate and plant hosts observed in various ecosystems may be due to seasonal variation and spatial heterogeneity across single time point studies (Kivlin and Hawkes, 2016). Thus, temporal sampling is needed to capture the seasonal dynamics of microbial communities..."

- "...Our research uses soil fungal measurements at a single point in time and there are likely to be many undetermined controlling factors, including seasonal variability and complex biological interactions. Despite this drawback, our approach allows us to infer the distribution of soil fungal communities and

diversity more simply and at a lesser cost, to help better understand the diversity and biogeography of soil fungi in different habitats. Thus, our approach shows promise and could complement molecular methods. We hope that our study will stimulate further research towards achieving more widespread characterisation of fungal abundance and diversity, which will help to deepen our understanding of fungal biology, biogeography and their environmental controls. Different spectra, new sensing technologies and improved methods could also improve the spectro-transfer approach..."

- We removed discussion that refers to soil health because we understand that this is a rather 'controversial' topic. Our research here is different and does not necessarily contribute to that discussion.

- We restructured and rewrote parts of the conclusions to emphasise that this work does not aim to provide a replacement method to measure soil fungi, but an encouraging new method that could help to complement the more expensive molecular method. The revised conclusions are: "Our study contributes to the development of methods that could complement, not replace, molecular approaches for characterising and better understanding the diversity and biogeography of soil fungi. We have shown that deep learning spectro-transfer functions are a promising new method for estimating soil fungal communities' relative abundance and diversity. The optimised 1D-CNNs outperformed the six other machine learning algorithms tested for estimating the relative abundance of fungal phyla and diversity. The spectro-transfer functions (with vis–NIR spectra and soil and environmental data) produced more accurate estimates ($R^2$ 0.45–0.73) than the spectroscopic models (only vis–NIR spectra; $R^2$ 0.36–0.55) and models with only the soil and environmental data ($R^2$ 0.38–0.60). As well as the soil organic and mineral composition, represented by vis–NIR spectra, other edaphic, climatic, and biotic factors including soil nutrients, pH, bulk density, potential evapotranspiration, the soil-water balance and net primary

productivity were important predictors in the modelling. "

**Authors:** Regarding the reviewer's other comments, we revised as follows:

- We clarified the distinction between woodlands and forests in the Method, as follows: "...Woodlands in Australia represent ecosystems which contain widely spaced trees, the crowns of which do not touch. Woodlands consist of areas with fewer and more scattered trees than forests. In temperate Australia, woodlands are mainly dominated by Eucalyptus species. Temperate woodlands occur predominantly in regions with a mean annual rainfall of between 250–800mm, forming a transitional zone between the higher rainfall forested margins of the continent and the shrub and grasslands of the arid interior....".

- We removed repetition and only present the relevant text in the Results: "In total, more than 60 million quality filtered sequences in the whole dataset were obtained, with an average of 107 310 sequences per sample. When we clustered the sequences at 97% similarity level 202200 OTUs were detected. Each sample had an average of 666 OTUs" .

- We revised the caption of Figure 1 to remind readers that the graph shows mean abundances: "...The mean relative abundances of dominant fungal phyla and unclassified "Others" taxa in five ecosystem types..."

- We added the SG1Der spectra in Fig. 2. We revised the relevant sections in the Methods and Results. Please see above.

- Regarding the comment on the '...absence of synthetic performance indicators such as the ratio of performance to deviation or the ratio of performance to interquartile distance...' We have used evaluation metrics that quantify the error in the model estimates in terms of their inaccuracy (RMSE), bias (ME) and imprecision (SDE) (such that $RMSE^2 = ME^2 + SDE^2$) and the $R^2$. Reporting RPDs or RPIQs will not help to better characterise or compare the errors and would only be redundant.

**References**

Janssen, P. and Heuberger, P.: Calibration of process-oriented models, Ecological Modelling, 83, 55-66, 1995.

Breiman, L.: Random forests, Machine learning, 45, 5-32, 2001.

Fisher, A., Rudin, C., and Dominici, F.: All Models are Wrong, but Many are Useful: Learning a Variable's Importance by Studying an EntireClass of Prediction Models Simultaneously, Journal of Machine Learning Research, 20, 1-81, 2019.

Delgado-Baquerizo, M., Oliverio, A. M., Brewer, T. E., et al, 2018. A global atlas of the dominant bacteria found in soil. Science, 359 (6373), 320-325.

Duan, Y, Xie N, Song, Z,et al., 2018. A High-Resolution Time Series Reveals Distinct Seasonal Patterns of Planktonic Fungi at a Temperate Coastal Ocean Site (Beaufort, North Carolina, USA). Appl Environ Microbiol., 84(21), e00967-18.

Hart, M. M., Cross, A. T., D'Agui, H. M.,et al., 2020.Examining assumptions of soil microbial ecology in the monitoring of ecological restoration, Ecological Solutions and Evidence, 1, e12031.

Kivlin, S. N., Hawkes, C. V., 2016. Tree species, spatial heterogeneity, and seasonality drive soil fungal abundance, richness, and composition in Neotropical rainforests, Environmental Microbiology, 18, 4662–4673, 201.

Talley, S.M., Coley, P.D., Kursar, T.A. 2002. The effects of weather on fungal abundance and richness among 25 communities in the Intermountain West. BMC Ecol., 2, 7.

---

## Author Response (AR2)

**Response to reviewers: Estimating soil fungal abundance and diversity at a macroecological scale with deep learning spectrotransfer functions by Yang et al.**

We have addressed the topical editor's comment. Our response indicating the changes we made in our revision is in (in blue text and preceded by **Authors:**).

**Topical Editor: Comments to the author**

I would like to thank you for the new version and the changes you made. The manuscript has been improved after addressing the comments from reviewers. However, I have a final comment. I did not find any discussion about the ranking of the prediction performances for the different soil fungal abundances. Figure 3 shows the different R2 but why Mortierellomycota and Mucoromycota produced the largest R2 while ACE produced the smallest ones? I think we need a paragraph in the discussion about this results ?

**Authors:** Regarding the predictability of the relative abundance of fungal phyla and diversity, we have revised the manuscript, as suggested by the editor, by including a new paragraph in the Discussion section (around lines 225–235) as follows: "...The spectro-transfer functions explained less than 60% of the variance in the two dominant phyla, the Ascomycota and Basidiomycota, representing 80% of the total fungal relative abundance. In comparison, the spectro-transfer functions could explain more than 70% of the variance in the Mortierellomycota and Mucoromycota, which were less abundant in soil. The reason for the different predictability might be the coarse phylum-level identity. Compared with the Mortierellomycota and Mucoromycotawere, the Ascomycota and Basidiomycota are more complex phylogenetic classifications and consist of more diverse taxa with different phenotypic traits. These taxa have distinct ecological functions and environmental preferences, which might have reduced the predictability of their relative abundance at the

phylum level. Classifying taxa with similar habitat preferences or studying at a finer taxonomic resolution might provide better predictability and understanding of soil fungal communities. The spectro-transfer function for the ACE index could only explain around 50% of the variance in diversity. The reason might be that local geography, environmental conditions, and difficult-to-proxy long-term natural selection and evolution affect community diversity."

**Authors:** We also added a final sentence to the Conclusions (around line 320) as follows: "...We hope that our study will provide food-for-thought for further research on the measurement and estimation of fungal abundance and diversity. We believe that improvements will be possible as new technologies and methodologies develop that will also help to deepen our understanding of fungal biology and biogeography."